# Inverse Learning of Symmetries

**Mario Wieser**[1,2], **Sonali Parbhoo**[3], **Aleksander Wieczorek**[1,2], **Volker Roth**[1,2]
[1]University of Basel, Switzerland
[2]National Centre for Computational Design and Discovery of Novel Materials MARVEL,
University of Basel, Switzerland
[3]John A. Paulson School of Engineering and Applied Sciences, Harvard University, USA
`mario.wieser@unibas.ch`

## Abstract

Symmetry transformations induce invariances which are frequently described with deep latent variable models. In many complex domains, such as the chemical space, invariances can be observed, yet the corresponding symmetry transformation cannot be formulated analytically. We propose to learn the symmetry transformation with a model consisting of two latent subspaces, where the first subspace captures the target and the second subspace the remaining invariant information. Our approach is based on the deep information bottleneck in combination with a continuous mutual information regulariser. Unlike previous methods, we focus on the challenging task of minimising mutual information in continuous domains. To this end, we base the calculation of mutual information on correlation matrices in combination with a bijective variable transformation. Extensive experiments demonstrate that our model outperforms state-of-the-art methods on artificial and molecular datasets.

## 1 Introduction

In physics, symmetries are used to model quantities which are retained after applying a certain class of transformations. From the mathematical perspective, symmetry can be seen as an invariance property of mappings, where such mappings leave a variable unchanged. Consider the example of rotational invariance from Figure 1a. We first observe the 3D representation of a specific molecule $m$. The molecule is then rotated. For any rotation $g$, we calculate the distance matrix $D$ between the atoms of the rotated molecule $g(m)$ with a predefined function $f$. Note that a rotation is a simple transformation which admits a straightforward analytical form. As $g$ induces an invariance class, we obtain the same distance matrix for every rotation $g$, i.e. $f(m) = f(g(m))$ for any rotation $g$.

Now, consider highly complex domains e.g. the chemical space, where analytical forms of symmetry transformations $g$ are difficult or impossible to find (Figure 1b). The task of discovering novel molecules for the design of organic solar cells in material science is an example of such a domain. Here, all molecules must possess specific properties, e.g. a bandgap energy of exactly 1.4 eV [37], in order to adequately generate electricity from the solar spectrum. In such scenarios, no predefined symmetry transformation (such as rotation) is known or can be assumed. For example, there exist various discrete molecular graphs with different atom and bond composition that result in the equivalent band gap energy. The only available data defining our invariance class are the $\{m, e\}^n$ numeric point-wise samples from the function $f$ where $n$ is the number of samples, $m$ the molecule and $e = f(m)$ the bandgap energy. Therefore, no analytical form of a symmetry transformation $g$ which alters the molecule $m$ and leaves the bandgap energy $e$ unchanged can be assumed.

The goal of our model is thus to learn the class of symmetry transformations $g$ which result in a symmetry property $f$ of the modelled system. To this end, we learn a continuous data representation and the corresponding continuous symmetry transformation in an inverse fashion from data samples $\{m, e\}^n$ only. To do so, we introduce the Symmetry-Transformation Information Bottleneck (STIB)

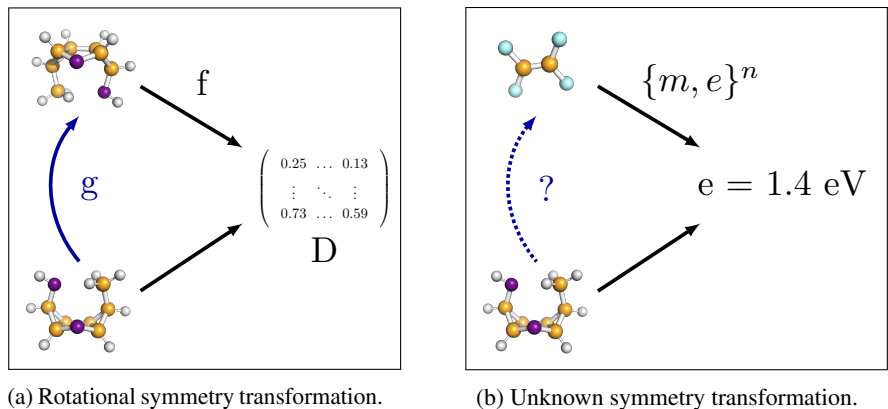

(a) Rotational symmetry transformation.　　(b) Unknown symmetry transformation.

Figure 1: **Left:** a molecule is rotated by $g$ admitting an analytical form. The distance matrix $D$ between atoms is calculated by a known function $f$ and remains unchanged for all rotations. **Right:** $n$ samples $\{m, e\}^n$ where $m$ is the molecule and $e$ the bandgap energy. These samples approximate the function $f$ whereas the class of functions $g$ leading to the same bandgap energy is unknown.

where we encode the input $X$ (e.g. a molecule) into a latent space $Z$ and subsequently decode it to $X$ and a preselected target property $Y$ (e.g. the bandgap energy). Specifically, we divide the latent space into two subspaces $Z_0$ and $Z_1$ to explore the variations of the data with respect to a specific target. Here, $Z_1$ is the subspace that contains information about input and target, while $Z_0$ is the subspace that is invariant to the target. In doing so, we capture symmetry transformations not affecting the target $Y$ in the isolated latent space $Z_0$.

The central element of STIB is minimising the information about continuous $Y$ (e.g. bandgap energy) present in $Z_0$ by employing adversarial learning. In contrast, cognate models have to the best of our knowledge solely focused on discrete $Y$. The potential reason is that naively using the negative log-likelihood (NLL) as done for maximising mutual information in other deep information bottleneck models leads to critical problems in continuous domains. This stems from the fact that fundamental properties of mutual information, such as invariance to one-to-one transformations, are not captured by this mutual information estimator. Simple alternatives such as employing a coarse-grained discretisation approach as proposed in [35] are not feasible in our complex domain. The main reason is that we want to consider multiple properties at once, every one of which might require a high-resolution. Simultaneous high-resolutional discretisation of multiple targets would result in an intractable classification problem.

To overcome the aforementioned issues, we propose a new loss function based on Gaussian mutual information with a bijective variable transformation as an addition to our modelling approach. In contrast to using the NLL, this enables the calculation of the full mutual information on the basis of correlations. Thus, we ensure mutual information estimation that is invariant against linear one-to-one transformations. In summary, we make the following contributions:

1. We introduce a deep information bottleneck model that learns a continuous low-dimensional representation of the input data. We augment it with an adversarial training mechanism and a partitioned latent space to learn symmetry transformations based on this representation.

2. We further propose a continuous mutual information regulation approach based on correlation matrices. This makes it possible to address the issue of one-to-one transformations in the continuous domain.

3. Experiments on an artificial as well as two molecular datasets demonstrate that the proposed model learns both pre-defined and arbitrary symmetry transformations and outperforms state-of-the-art methods.

## 2　Related Work

**Information Bottleneck and its connections.**　The Information Bottleneck (IB) method [40] describes an information theoretic approach to compressing a random variable $X$ with respect to a

second random variable $Y$. The standard formulation of the IB uses only discrete random variables. However, various relaxations such as for Gaussian [4] and meta-Gaussian variables [32], have also been proposed. In addition, a deep latent variable formulation of the IB [2, 1, 42] method and subsequent applications for causality [30, 29] or archetypal analysis [15, 16] have been introduced.

**Enforcing invariance in latent representations.** Bouchacourt et al. [3] introduced a multi-level variational autoencoder (VAE). Here, the latent space is decomposed into a local feature space that is only relevant for a subgroup and a global feature space. A more common technique to introduce invariance makes use of adversarial networks [12]. Specifically, the idea is to combine VAEs and GANs, where the discriminator tries to predict attributes, and the encoder network tries to prevent this [8, 23]. Perhaps most closely related to this study is the work of [21] where the authors propose a mutual information regulariser to learn isolated subspaces for binary targets. However, these approaches are only applicable for discrete attributes and our work tackles the more fundamental and challenging problem of learning symmetry transformations for continuous properties.

**Relations to Fairness.** Our work has close connections to fairness. Here, the main idea is to penalise the model for presence of nuisance factors $S$ that have an unintended influence on the prediction $Y$ to archive better predictions. Louzios *et al.* [25], for example, developed a fairness constraint for the latent space based on maximum mean discrepancy (MMD) to become invariant to nuisance variables. Later, Xie *et al.* [43] proposed an adversarial approach to become invariant against nuisance factors $S$. In addition, Moyer *et al.* [28] introduced a novel objective to overcome the disadvantages of adversarial training. Subsequently, Jaiswal *et al.* [14] built on these methods by introducing a regularisation scheme based on the LSTM [13] forget mechanism. In contrast to the described ideas, our work focuses on learning a symmetry transformation for continuous $Y$ instead of removing nuisance factors $S$. Furthermore, we are interested in learning a generative model instead of solely improving downstream predictions.

**Connections to Disentanglement.** Another important line of research denotes disentanglement of latent factors of variation. Higgins et al. citebeta introduced an additional Lagrange parameter to steer the disentanglement process in VAEs. Chen et al. [6] introduced a mutual information regulariser to learn disentangled representations in GANs and Mathieu et al. [27] considered combining VAEs with GANs. Other important work in this direction include [5, 17, 35, 9, 39].

## 3 Preliminaries

### 3.1 Deep Information Bottleneck

The Deep Variational Information Bottleneck (DVIB) [2] is a compression technique based on mutual information. The main goal is to compress a random variable $X$ into a random variable $Z$ while retaining side information about a third random variable $Y$. Note that DVIB builds on VAE [18, 34], in that the mutual information terms in the former are equivalent to the VAE encoder $q(z|x)$ and decoder $p(x|z)$ [2]. Therefore, the VAE remains a special case of DVIB where the compression parameter $\lambda$ is set to 1 and $Y$ is replaced by the input $X$ in $I(Z; Y)$. $I$ represents the mutual information between two random variables. Achieving the optimal compression requires solving the following parametric problem:

$$\min_{\phi, \theta} I_\phi(Z; X) - \lambda I_{\phi, \theta}(Z; Y), \tag{1}$$

where we assume a parametric form of the conditionals $p_\phi(z|x)$ and $p_\theta(y|z)$. $\phi, \theta$ represent the neural network parameters and $\lambda$ controls the degree of compression.

The mutual information terms can be expressed as:

$$I(Z; X) = \mathbb{E}_{p(x)} D_{KL}(p_\phi(z|x) \| p(z)) \tag{2}$$

$$I(Z; Y) \geq \mathbb{E}_{p(x,y)} \mathbb{E}_{p_\phi(z|x)} \log p_\theta(y|z) + h(Y) \tag{3}$$

where $D_{KL}$ denotes the Kullback-Leibler divergence and $\mathbb{E}$ the expectation. For the details on the last inequality in Eq. (3), see [41].

## 3.2 Adversarial Information Elimination

A common approach to remove information from latent representations in the context of VAEs is using adversarial training [23, 8, 21]. The main idea is to train an auxiliary network $a_\psi(z)$ which tries to correctly predict an output $b$ from the latent representation $z$ by minimising the classification error. Concurrently, an adversary, in our case the encoder network of the VAE, tries to prevent this. To this end, the encoder $q_\theta(z|x)$ attempts to generate adversarial representations $z$ which contain no information about $b$ by maximising the loss $\mathcal{L}$ with respect to parameters $\theta$. The overall problem may then be expressed as an adversarial game where we compute:

$$\max_\theta \min_\psi \mathcal{L}(a_\psi(p_\theta(z|x)), b), \tag{4}$$

with $\mathcal{L}$ denoting the cross-entropy loss. While this approach is applicable for discrete domains, generalising this loss function to continuous settings can lead to severe problems in practice. We elaborate on this issue in the next section.

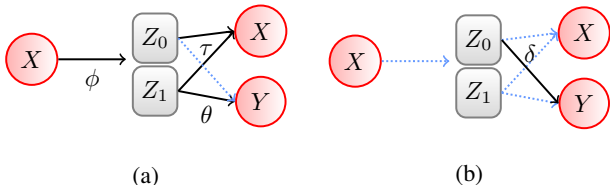

(a)                                              (b)

Figure 2: Graphical illustration of our two-step adversarial training approach. Red circles denote observed input/output of our model. Gray rectangles represent the latent representation which is divided into two separate subspaces $Z_0$ and $Z_1$. Blue dotted arrows represent neural networks with fixed parameters Black arrows describe neural networks with trainable parameters. Greek letters define neural network parameters. In the first step (Figure 2a), we try to learn a representation of $Z_0$ which minimises the mutual information between $Z_0$ and $Y$ by updating $\phi, \theta$ and $\tau$. In the adversary step (Figure 2b), we maximise the mutual information between $Z_0$ and $Y$ by updating $\delta$.

# 4 Symmetry-Invariant Information Bottleneck

As previously described in Section 1, our goal is to learn symmetry transformations $g$ based on observations $X$ and $Y$ (see Figure 1b). Here, $X$ and $Y$ may be complex objects, such as molecules and its corresponding bandgap energies, which are difficult to manipulate consistently. In order to overcome this issue, we aim to learn a continuous low-dimensional representation of our input data $X$ and $Y$ in Euclidian space. To do so, we augment the traditional deep information bottleneck formulation (Eq. (1)) with an additional decoder reconstructing $X$ from $Z$. Our base model is thus defined as an augmented parametric formulation of the information bottleneck (see Eq. (1)):

$$\min_{\phi, \theta, \tau} I_\phi(Z; X) - \lambda(I_{\phi,\theta}(Z; Y) + I_{\phi,\tau}(Z; X)), \tag{5}$$

where $\phi, \theta, \tau$ describe neural network parameters and $\lambda$ the compression factor.

**Theoretical Concept.** Based on the model specified in Eq. (5), we provide a novel approach to learn symmetry transformations on latent representations $Z$. To this end, we propose partitioning the latent space $Z$ into two components, $Z_0$ and $Z_1$. $Z_1$ is intended to capture all information about $Y$, while $Z_0$ should contain all remaining information about $X$. That is, changing $Z_0$ should change $X$ but *not affect* the value of $Y$. Thus, $Z_0$ expresses a surrogate of the unknown function $g$ which is depicted in Figure 1b. However, simply partitioning the latent space is not sufficient, since information about $Y$ may still be encoded in $Z_0$.

To address this task, we propose combining the model defined in Eq. (5) with an adversarial approach (Section 3.2). The resulting model thus reduces to playing an adversarial game of minimising and maximising the mutual information $I_\delta(Z_0, Y)$ where $\delta$ denotes the neural network parameters. This ensures that $Z_0$ contains no information about $Y$. In more detail, our approach is formulated as follows:

In the first step (see Figure 2a), our model learns a low-dimensional representation $Z_0$ and $Z_1$ of $X$ and $Y$ by maximising the mutual information between $I_{\phi,\tau}(Z_0, Z_1; X)$ and $I_{\phi,\theta}(Z_1; Y)$. At the same time, our algorithm tries to eliminate information about $Y$ from $Z_0$ by minimising the mutual $I_\delta(Z_0; Y)$ via changing $Z_0$ with fixed parameters $\delta$ (brown part of Eq. 6).

$$\mathcal{L}_1 = \min_{\phi,\theta,\tau} I_\phi(X;Z) - \lambda\Big(I_{\phi,\tau}(Z_0, Z_1; X) + I_{\phi,\theta}(Z_1; Y) - I_\delta(Z_0; Y)\Big) \tag{6}$$

The second step defines the adversary of our model and is illustrated in Figure 2b. Here, we try to maximise the mutual information $I_\delta(Z_0; Y)$ (purple part of Eq. 7) given the current representation of $Z_0$. To do so, we fix all model parameters except of $\delta$ and update the parameters accordingly.

$$\mathcal{L}_2 = \min_{\delta} I_\phi(X;Z) - \lambda\Big(I_{\phi,\tau}(Z_0, Z_1; X) + I_{\phi,\theta}(Z_1; Y) + I_\delta(Z_0; Y)\Big) \tag{7}$$

The resulting loss functions $\mathcal{L}_1$ and $\mathcal{L}_2$ are alternately optimised until convergence. Yet, minimising mutual information for continuous variables, such as bandgap energy, remains a challenging task.

**Challenges in Continuous Domains.** Mutual information is invariant against the class of one-to-one transformations as it depends only on the copula [26]. That is, $I(X; Y) = I(f(X); g(Y))$, where $g$ and $f$ denote one-to-one transformations. Related models extending the deep information bottleneck define mutual information for random variables as the NLL plus marginal entropy (see Eq. (3)). Building on this, only the NLL part of mutual information is optimised. This part alone is, however, not invariant against such transformations. This gives rise to problems in tasks involving minimising mutual information, e.g. as required by our adversary in Eq. (6). This is because while maximising the NLL, the network can learn solutions stemming from one-to-one transformations of the marginal. For example, the network might maximise the NLL by adding only a large bias to the output. This leads to an increased NLL even though MI remains unchanged. This results in solutions which are not desired when minimising MI. Therefore, we require a more sophisticated approach that estimates the full mutual information and hence introduces invariance against one-to-one transformations.

**Suggested Solution.** We propose a to estimate the Gaussian mutual information based on the correlation matrices and thus circumvent the problems discussed in the previous paragraph. This is because mutual information based on correlations can be meaningfully minimised, as it is by definition invariant against one-to-one transformations. In the next paragraph we demonstrate how we relax the Gaussian assumption. Mutual information can be decomposed into a sum of *multi-informations* [24]:

$$I(Z_0; Y) = M(Z_0; Y) - M(Z_0) - M(Y), \tag{8}$$

where the specific multi-information terms for Gaussian variables have the following forms:

$$M(Z_0, Y) = \frac{1}{2}\log\left((2\pi e)^{n+m}|R_{Z_0 Y}|\right),$$

$$M(Z_0) = \frac{1}{2}\log\left((2\pi e)^n|R_{Z_0}|\right),$$

$$M(Y) = \frac{1}{2}\log\left((2\pi e)^m|R_Y|\right), \tag{9}$$

where $Z_0$ and $Y$ are $n$- and $m$-dimensional, respectively. $R_{Z_0 Y}$, $R_{Z_0}$ and $R_Y$ denote the sample covariance matrices of $Z_0 Y$, $Z_0$ and $Y$, respectively. In practice, we calculate the correlation matrices based on the sample covariance. Note that in the Gaussian setting in which our model is defined, correlation is defined as a deterministic function to the mutual information [7].

**Relaxing the Gaussian Assumption.** As previously stated, to deal with probabilistic models, we require a simple parametric model. Hence, we made the strong parametric assumptions that both $Z_0$ and $Y$ are Gaussian distributed. However, the target variable $Y$ does not necessarily follow a Gaussian distribution. Our approach to relax this assumption is to open the limited Gaussian distribution to the class of all non-linearly transformed Gaussian distributions. For this reason, we equip the model with a proxy bijective mapping $Y \leftrightarrow \tilde{Y}$ (Figure 3) to introduce more flexibility. This mapping is implemented as two additional networks between $Y$ and a new proxy variable $\tilde{Y}$. The parameters are added to the existing parameters $\delta$. We subsequently treat $\tilde{Y}$ as values to be predicted from $Y$.

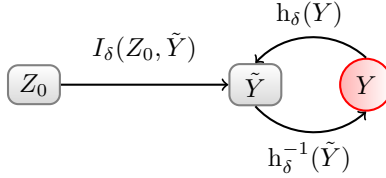

Figure 3: Model extended with the bijective mapping between $Y$ and $\tilde{Y}$. Solid arrows depict a nonlinear function parametrised by a neural network. Gray rectangles denote latent and red circles observed variables.

We found in our experiments that this approximate approach is sufficient to ensure the invariance property. Our approach makes it possible to compute the mutual information between $Z_0$ and $Y$ (or its proxy, $\tilde{Y}$) analytically with the formula for Gaussian variables. Thus, we augment $\mathcal{L}_2$ as follows:

$$\mathcal{L}_{\text{bijection}} = \mathcal{L}_2 + \beta\|h_\delta^{-1}(h_\delta(Y)) - Y\|_2^2$$

## 5 Experiments

A description of setups and additional experiments can be found in the supplementary materials[1].

### 5.1 Artificial Experiments

**Dataset.** Our dataset is generated as follows: Our input consists of two input vectors $x_0$ and $x_1$. Here, the input vectors are drawn from a uniform distribution defined on $[0, 1]$ and further multiplied by 8 and subtracted by 4. All input vectors form our input matrix $X$. Subsequently, we define a latent variable $z_0$. Here, $z_0$ is calculated as $2x_0 + x_1 + 10^{-1}\epsilon$ where $\epsilon \sim \mathcal{N}(0, 1)$. Last, we calculate two target variables $y_0$ and $y_1$. In doing so, $y_0$ is calculated by $5 \cdot 10^{-2}(z_0 + bz_0)\cos(z_0) + 2 \cdot 10^{-1}\epsilon_1$ where $\epsilon_1 \sim \mathcal{N}(0, 1)$ and $b$ is 10. $y_1$ is defined as $5 \cdot 10^{-2}(z_0 + bz_0)\sin(z_0) + 2 \cdot 10^{-1}\epsilon_2$ with $\epsilon_2 \sim \mathcal{N}(0, 1)$. Thus, $y_0$ and $y_1$ form a spiral where the angle and the radius are highly correlated. For visualisation purposes, we bin and colour code the values of $Y$ in the following experiments. During the training and testing phase, samples are drawn from this data generation process.

**Experiment 1. Examining the Latent Space** We demonstrate the ability of our model to learn a symmetry transformation which admits an analytical form from observations only. We compare our method with VAE [11] and STIB without regulariser for this purpose. Here, we use the same latent space structure that was described in the experimental setup. Subsequently, we plot the first dimension of $Z_0$ (x-axis) against $Z_1$ (y-axis) for all three methods. Due to the fact that every dimension in the VAE model contains information about the target, we plotted the first against the second dimension for simplicity. The horizontal coloured lines indicate that our approach (Fig. 4c) is able to learn a well defined symmetry transformation, because changing the value of the x-axis does not change the target $Y$. In contrast, the VAE (Fig. 4a) and STIB without any regulariser (Fig. 4b) are not able to preserve this invariance property and encode information about $Y$ in $Z_0$ simultaneously. This can be clearly noted by the colour change of horizontal lines. That is, modifying the invariant space would still result in a change of $Y$ and thus requires our mutual information regulariser.

**Experiment 2. Quantitative Evaluation** Here, we provide a quantitative comparison study with five different models in order to demonstrate the impact of our novel model architecture and mutual information regulariser. In addition to the models considered in Experiment 1, we compare to conditional VAE (CVAE) [38] and conditional Information Bottleneck (CVIB) [28] in Table 1. The setup is identical as described in the experimental setup (see Supplement). We compare the reconstruction MAE of $X$ and $Y$ as well as the amount of information which is remaining in the invariant space by measuring the mutual information between $Z_0$ and $Y$. Our study shows that we are able to obtain competitive reconstruction results for both $X$ and $Y$ for all of the models. However, we encounter a large difference between the models with respect to the remaining target information $Y$ in the latent space $Z_0$. In order to quantify the differences, we calculated the mutual information

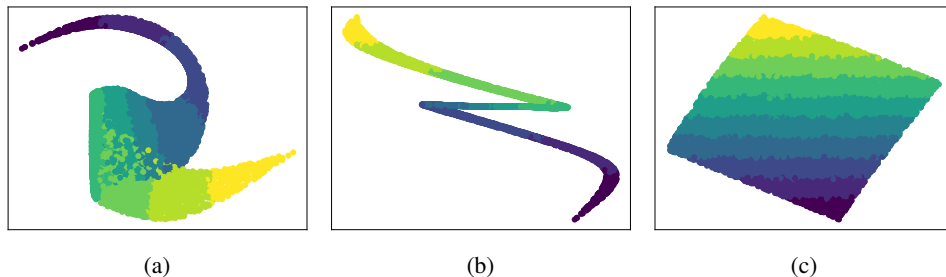

| (a) | (b) | (c) |

Figure 4: Figure 4a depicts the latent space of VAE where the first two dimensions are plotted. In contrast, Figure 4b shows the latent space of STIB that was trained without our regulariser. Here, the first invariant dimension $Z_0$ (x-axis) is plotted against the dimension of $Z_1$ (y-axis). Figure 4c illustrates first dimension of the invariant latent space $Z_0$ (x-axis) plotted against $Z_1$ (y-axis) after being trained by our method. Horizontal coloured lines in the bottom right panel indicate invariant with respect to the target $Y$. In remaining panels the stripe structure is broken. Black arrows denote the invariant direction.

using a nonparametric Kraskov estimator [22] to obtain a consistent estimate for all the models we compared. We specifically avoid using the Gaussian mutual information in our comparisons here, because in the other models the (non-transformed) $Y$ is not necessarily Gaussian. Otherwise, we would end up with inaccurate mutual information estimates that make fair comparison infeasible. By using the Kraskov estimator, we observe that all competing models, $Z_0$ contain a large amount of mutual information about $Y$. In the VAE case, we obtain a mutual information if 3.89 bits and with our method without regularisation a value of 3.85 bits. Moreover, CVAE and CVIB still contain 2.57 bits and 2.44 bits mutual information, respectively. However, if we employ our adversarial regulariser, we are able to decrease the mutual information to 0.30 bits. That is, we have approximately removed all information about $Y$ from $Z_0$. These quantitative results showcase the effectiveness of our method and support the qualitative results illustrated in Figure 4.

Table 1: Quantitative summary of results from artificial and QM9 experiments. Here, we consider the VAE, STIB without regularization, CVAE, CVIB, STIB. For all models in the artificial experiments the MAE reconstruction errors for $X$ and $Y$ are considered as well as the mutual information (MI) in bits between the invariant space $Z_0$ and $Y$ based on a Kraskov estimator. For the QM9 experiment, we consider accuracy for SMILES and MAE reconstruction errors for bandgap energy (gap) in eV, polarizability (pol) in bohr$^3$ as well as MI. Lower MAE and MI is better. STIB outperforms each of the baselines considered.

| MODEL | ARTIFICIAL EXPERIMENT | | | QM9 | | | |
|---|---|---|---|---|---|---|---|
| | MAE(X) | MAE(Y) | $\text{MI}_K(Z_0,Y)$ | SMILES | GAP | POL | $\text{MI}_K(Z_0,Y)$ |
| VAE | 0.21 | 0.49 | 3.89 | **0.98** | 0.28 | 0.75 | 1.54 |
| STIB W/O ADV. | **0.01** | 0.65 | 3.19 | **0.98** | 0.28 | **0.70** | 0.66 |
| CVAE | 0.33 | - | 2.57 | 0.91 | - | - | 0.56 |
| CVIB | 0.67 | - | 2.45 | 0.76 | - | - | **0.03** |
| STIB | 0.04 | **0.47** | **0.25** | **0.98** | **0.27** | 0.77 | 0.09 |

## 5.2 Real Experiment: Small Organic Molecules

**Dataset.** We use the 134K organic molecules from the QM9 database [31], which consists of up to nine main group atoms (C, O, N and F), not counting hydrogens. The chemical space of QM9 is based on the work of GDB-17 [36], as the largest virtual database of compounds to date, enumerating 166.4 billion molecules of up to 17 atoms of C, N, O, S, and halogens. GDB-17 is systematically generated using molecular connectivity graphs, and represents an attempt of complete and unbiased enumeration of the space of chemical compounds with small and stable organic molecules. Each molecule in QM9 has corresponding geometric, energetic, electronic and thermodynamic properties that are computed from Density Functional Theory (DFT) calculations. In all our experiments, we

focus on a subset of this dataset with a fixed stoichiometry ($C_7O_2H_{10}$), consisting of 6095 molecules and corresponding bandgap energy and polarisability as invariant properties. By restricting chemical space to fixed atomic composition we can focus on how changes in chemical bonds govern these properties.

**Experiment 3. Examining the Latent Space** Here, we demonstrate that our model can generalise to more than one target (see Experiment 1), meaning novel materials have to satisfy multiple properties and at the same time have a structural invariant subspace. We train a model with a subspace ($Z_0$) which contains no information about the molecular properties, bandgap energy and polarisability. In order to illustrate this relationship, we plot the first two dimensions of the property space $Z_1$ and colour coded points according to intervals for bandgap energy and polarisability (Figure 5a and Figure 5b respectively). The colour bins are equally spaced by the property range, where the minimum is 1.02 eV / 6.31 bohr$^3$ and the maximum is 16.93 eV / 143.43 bohr$^3$ for bandgap energies and polarisability, respectively. For simplicity and readability, we divide the invariant latent space $Z_0$ into ten sections and cumulatively group the points. Four sections were chosen for Figure 5. We note that binning is not necessary, but increases the readability of the figure. In every $Z_0$-section, we observe the stripe structure which means that $Z_0$ is invariant with respect to the target. In contrast, if $Z_0$ encoded any property information, the stripe structure would be broken as demonstrated in Panels 4a and 4b. Thus, our experiment clearly indicates that there is no change in the latent space structure with respect to bandgap energy and polarisability. Therefore, varying $Z_0$ will not affect the properties of the molecule.

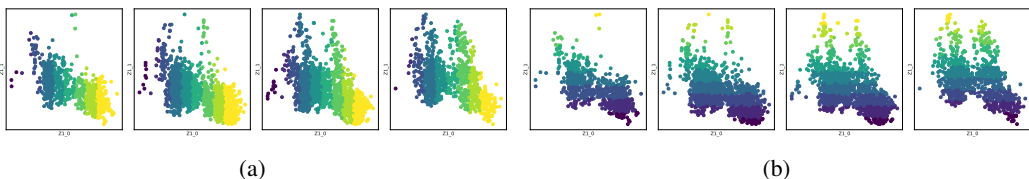

(a)                                            (b)

Figure 5: Latent space plots for the first two dimensions in property dependent $Z_1$. Colours illustrates binned target properties, bandgap energies (Fig. 5a) and polarisabilities (Fig. 5b). The bins are equally spaced by the property range. The values lie between 1.02 eV / 6.31 bohr$^3$ and 16.93 eV / 143.43 bohr$^3$ for bandgap energies and polarisability, respectively. The four figures for each property denote four binned sections along the property invariant dimension $Z_0$, out of a total of ten sections. The invariance is illustrated by the lack of overlap of the colour patterns for each section in $Z_0$.

**Experiment 4. Quantitative Evaluation** In this experiment, we perform a quantitative study on real data to demonstrate the effectiveness of our approach. We compare the baseline VAE, CVAE, CVIB, STIB without mutual information regularisation of the latent space and STIB with mutal information regularization (Table 1). Upon comparing the accuracy of correctly reconstructed SMILES strings and the MAE of the bandgap energy and polarisability, we obtain competitive reconstruction results. For all the models considered in the quantitative study, we obtained a SMILES reconstruction accuracy of 98% for VAE, 98%, for STIB without the adversarial training scheme, 91% for CVAE, 76% for CVIB and 98% for STIB. In addition, the bandgap and polarisability MAE for the VAE is 0.28 eV and 0.75 bohr$^3$, respectively. In comparison, the STIB without adversary reaches a bandgap error of 0.28 eV and a polarisability error of 0.70 bohr$^3$. Moreover, STIB obtains a MAE for bandgap energy of 0.27 eV and 0.77 bohr$^3$ for polarisability. This shows that our approach provides competitive results in both reconstruction tasks in comparison to the baseline. As previously described in Experiment 3, we additionally calculated the mutual information with a Kraskov estimator between the target-invariant space $Z_0$ and the target $Y$. In order to get a better estimate, we estimated the mutual information on the whole dataset. For both the baseline and our model without regularisation, we received a mutual information of 1.54 bits and 0.66 bits, respectively. Here, 1.54 bits represent the amount of mutual information if the entire $Z$ is considered (e.g. VAE). In addition, CVAE contains 0.56 bits mutual information. That implies that $Z_0$ still contains half the information about $Y$, whereas if we employ our regulariser, the mutual information is 0.09 bits. These quantitative results showcase that only STIB is able to approximatly remove all property information from $Z_0$ for real world applications and support the qualitative results obtained in Experiment 6. CVIB resulted in a slightly better MI estimate with 0.03 bits, however it has a much weaker reconstruction accuracy.

**Experiment 5: Generative Evaluation** In the last experiment, we investigate the generative nature of our model by inspecting the consistency of the property predictions. To do so, we explore two different points in $Z_1$ space, according to two different reference molecules, as seen in Figure 6a. The points in $Z_1$ space represent bandgap energies and polarisabilities of 6.00 eV / 76.46 bohr$^3$, and 8.22 eV / 74.76 bohr$^3$, (Figure 6a), respectively. Subsequently, we randomly sample from $Z_0$ to generate novel molecules as SMILES. Invalid SMILES that are not within the same stoichiometry are filtered out. The two generated molecules with the shortest latent space distance to the reference molecules are selected and depicted in Figure 6b. To demonstrate that $Z_0$ is property invariant, we perform a self-consistency check of the properties. To this end, we predict the properties of the generated molecules with our model and check if the properties lie within the expected error range (Table 1). We averaged the predicted properties over all generated molecules from the two reference points (approx. 37 molecules per point) and illustrated them as boxplots in Figure 6c. As the boxes are within the shaded background (error range), our experiment demonstrates the generative capabilities of the network by generating chemically correct novel molecules, which has self-consistent properties and hence learns property-invariant space $Z_0$ within the expected error of the model.

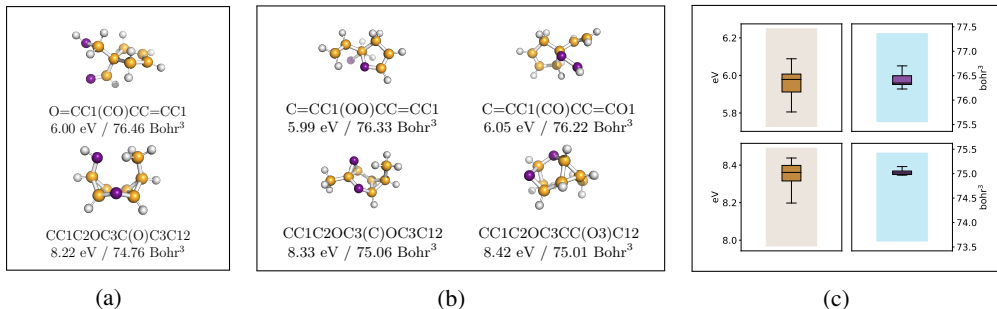

| (a) | (b) | (c) |
|---|---|---|

Figure 6: Illustrating the generative process of our model. The experiments for two different molecules are located row-wise. Figure 6a show the reference molecules which serve as our starting point with their corresponding properties. In Figure 6b, we plotted two generated molecules which are closest to the reference molecule. The properties from the generated molecules are estimated by using the prediction network of our model. Additionally, we predict the properties of all generated molecules (approx. 37 per point) and depict them as a box plot in Figure 6c, where the left box plot denotes the band gap energy and the right box plot the polarisability. The cross shaded background is the error confidence interval of our model.

# 6 Conclusion

Symmetry transformations constitute a central building block for a large number of machine learning algorithms. In simple cases, symmetry transformations can be formulated analytically, e.g. for rotation or translation (Figure 1a). However, in many complex domains, such as the chemical space, invariances can only be learned from data, for example when different molecules have the same bandgap energy (Figure 1b). In the latter scenario, the corresponding symmetry transformation cannot be formulated analytically. Hence, learning such symmetry transformations from observed data remains a highly relevant yet challenging task, for instance in drug discovery. To address this task, we make three distinct contributions:

1. We present *STIB*, that learns arbitrary continuous symmetry transformations from observations alone via adversarial training and a partitioned latent space.

2. In addition to our modelling contribution, we provide a technical solution for continuous mutual information estimation based on correlation matrices.

3. Experiments on an artificial as well as two molecular datasets show that the proposed model learns symmetry transformations for both well-defined and arbitrary functions, and outperforms state-of-the-art methods.

For future work, we would like to employ invertible networks [33, 20, 10, 19] to relax the Gaussian assumption for estimating the mutual information.

# 7 Broader Impact

Our society is currently faced with various existential threats such as climate change and fast spreading infectious diseases (Covid-19) that require immediate solutions. Many of these solutions, such as organic solar cells or drugs rely on the discovery of novel molecules. However, designing such molecules manually is heavily time-consuming as they have to fulfill specific properties. To overcome this limitation, we introduced a method to explore and generate candidate molecules that drastically speed up the discovery process. Despite the fact that our method constitutes an important contribution to tackle the described challenges, there might also be negative consequences. A potential downside is that drugs or techniques which rely on this method might not be available to individuals in developing countries for economic reasons. Therefore, it is crucial for us as a community as well as a society to monitor the usage of such models and correct potential deficiencies.

## Acknowledgements

We would like to thank Anders S. Christensen, Felix A. Faber, Puck van Gerwen, O. Anatole von Lilienfeld, Sebastian M. Keller, Jimmy C. Kromann, Vitali Nesterov and Maxim Samarin for insightful comments and helpful discussions, and thank Jack Klys and Jake Snell for sharing their code with us. Mario Wieser is partially supported by the NCCR MARVEL and grant 51MRP0158328 (SystemsX.ch HIV-X) funded by the Swiss National Science Foundation. Sonali Parbhoo is supported by the Swiss National Science Foundation project P2BSP2 184359 and NIH1R56MH115187. Aleksander Wieczorek is partially supported by grant CR32I2159682 funded by the Swiss National Science Foundation.

## Footnotes

[1]Code is publicly available at https://github.com/bmda-unibas/InverseLearningOfSymmetries

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
