[Supplementary Material]

# Appendix: Inverse Learning of Symmetries

## 1  Model

### 1.1  Implementation

In this section, we describe how to implement the mutual information terms of our model in practice. To do so, we describe the encoder term $I(Z;X)$, which is calculated as the Kullback-Leibler divergence ($\mathrm{D}_{KL}$) between $p_\phi(z|x)$ and $p(z)$. We implement $p_\phi(z|x)$ as a Gaussian distribution with parameters $\mu$ and $\sigma$ that are learned by a neural network with parameters $\phi$. Subsequently, we define the second part of the KL divergence $p(z)$ to be a simple Gaussian prior $\mathcal{N}(0,1)$. Due to the fact that we have defined both parts as Gaussian distributions, it is thus possible to calculate the KL divergence analytically.

$$I(Z;X) = \mathbb{E}_{p(x)}D_{KL}(p_\phi(z|x)\|p(z)) \tag{1}$$

With the first mutual information term (Eq. 1), we learn a compressed representation $Z$ of the input $X$. In a next step, we partition $Z$ into two latent spaces $Z_0$ and $Z_1$. Here, we assume that $Z$ consists of $k$ latent dimensions where $Z_0$ contains $h = 0 \ldots d$ dimensions of $Z$ where $d < k$. $Z_1$ thus includes the remaining $m = d + 1 \ldots k$ dimensions. However upon this point, we have only learned the parameters of the Gaussian distribution. In order to sample from the latent space $Z$, we make use of the reparametrisation trick introduced in [5, 13]. To do so, a sample from $Z$ can be drawn by reformulating the Gaussian distribution. Therefore, we apply the following formulation: $\mu + \sigma \odot \epsilon$ where $\epsilon$ is drawn from $\mathcal{N}(0,1)$ and $\mu$ and $\sigma$ are the learned parameters.

In the last step, we describe how the decoders for $X, Y$ have been implemented by our model. The decoders employ the following form:

$$
\begin{aligned}
I(Z_0, Z_1; X) = &\ \mathbb{E}_{p(x)}\mathbb{E}_{\epsilon \sim \mathcal{N}(0,I)} \\
&\sum_j \log p_{\tau_j}(x_j | z = \mu_k(x) + \mathrm{diag}(\sigma_k(x)), \odot\epsilon),
\end{aligned}
$$

$$
\begin{aligned}
I(Z_1; Y) = &\ \mathbb{E}_{p(x,y)}\mathbb{E}_{\epsilon \sim \mathcal{N}(0,I)} \\
&\sum_j \log p_{\theta_j}(y_j | z_1 = \mu_h(x) + \mathrm{diag}(\sigma_h(x)), \odot\epsilon),
\end{aligned}
$$

$$
\begin{aligned}
I(Z_0; Y) = &\ \mathbb{E}_{p(x,y)}\mathbb{E}_{\epsilon \sim \mathcal{N}(0,I)} \\
&\sum_j \log p_{\delta}(y_j | z_0 = \mu_d(x) + \mathrm{diag}(\sigma_d(x)) \odot \epsilon),
\end{aligned}
$$

where $j$ denotes the j$^{\text{th}}$ data sample. The distributions $p_\tau(x|z)$, $p_\theta(y|z_1)$ and $p_\delta(y|z_0)$ are implemented as neural networks with parameters $\phi, \tau, \delta$. In contrast to the encoder, these distributions may be arbitrarily chosen as we do not make any assumption about their form.

## 1.2 Training Procedure

Our learning procedure is described in Algorithm 1. For every epoch, we sample $i$ minibatches, where the number of $i$ is determined by the batchsize. In the first part of our training algorithm (see Minimisation Step), we try to minimise the mutual information between $Z_0$ and $Y$. Therefore, we first encode $X$ in our latent $Z$ (line 6). Subsequently, we split $Z$ in $Z_0$ and $Z_1$. After, we decode $Z$ to $X$ and $Z_1$, $Z_0$ to $Y$ in lines 6-10, respectively. Consequently, we update the parameters in line 12 by employing the loss function in Eq. 6 (main paper). Having fixed the parameters $\delta$ we update the latent representation $Z_0$ such that it encodes minimal mutual information about $Y$.

The second step (see Maximisation Step) denotes the adversarial part of our model. Again we encode $X$ into $Z$ and partition the space to $Z_0$ and $Z_1$. In contrast to the minimisation step, we try to predict $Y$ from $Z_0$ by updating parameters $\delta$ while fixing the remaining neural network parameters $\phi, \theta$ and $\tau$ (see Eq. 7 in main paper). In order to relax the Gaussian assumption, we extend our model by two additional neural networks $h$ and $h^{-1}$ and add their parameters to the existing parameters $\delta$ in line 20.

As a last step, we increase the compression parameter $\lambda$ by a predefined factor $l$ (line 24) after every epoch. Finally, we can train the described adversarial algorithm by any gradient descent method until convergence.

---

**Algorithm 1** Symmetry-Transformation Information Bottleneck

---

**Input:** input $x$, target $y$
 1: **for** each epoch **do**
 2:     sample $i$ minibatches of $x$ and $y$
 3:     **for** each minibatch $i$ **do**
 4:
 5:         Minimisation Step
 6:         encode $x_i$ into $p_\phi(z_i \mid x_i)$
 7:         split $z_i$ in $z_0$ and $z_1$
 8:         decode $z_0$ and $z_1$ to obtain $p_\tau(x_i \mid z_0, z_1)$
 9:         decode $z_1$ to obtain $p_\theta(y_i \mid z_1)$
10:         decode $z_0$ to obtain $p_\delta(y_i \mid z_0)$
11:
12:         update $\phi, \theta, \tau$ by taking a gradient step
13:
14:         Maximisation Step
15:         encode $x_i$ into $p_\phi(z_i \mid x_i)$
16:         split $z_i$ in $z_0$ and $z_1$
17:         decode $z_0$ to obtain $p_\delta(y_i \mid z_0)$
18:
19:         Relaxing Gaussian Assumption
20:         decode $y$ from $h_\delta^{-1}(h_\delta(y))$
21:
22:         update $\delta$ by taking a gradient step
23:     **end for**
24:     increase $\lambda$ by factor $l$
25: **end for**

---

## 1.3 Other Methods for MI Estimation

Quantifying mutual information is essential in information bottleneck methods. The naive approach requires estimating the joint distribution of the variables. One way to overcome this requirement are non-parametric $k$-nearest-neighbours-based estimators [6, 12], but they require an exponential number of samples when the true mutual information is large [2]. A number of methods estimating lower bounds of mutual information exist [1, 11]. Such bounds, however, suffer from inherent statistical limitations [8]. In this paper, we make use of the analytic form of the Gaussian mutual information based on the correlation matrix and subsequently relax the Gaussian assumption with neural networks.

# 2 Experiments

## 2.1 Artificial Experiments

### 2.1.1 Experimental Setup:

**STIB** For our setup, our encoding network consists of two fully connected layers with 256 neurons without bias. This is followed by a latent layer with three nodes that models the means of our three dimensional latent space where the variances are modeled as free parameters. Here, we split $Z$ in a one-dimensional space $Z_1$ and two-dimensional space $Z_0$ which contain no information about $Y$. The decoder uses two neural networks with fully connected layers with 256 neurons for reconstructing the input and predicting the target. In addition, we model an adversarial network with two fully connected layers each with 256 neurons to predict the target from our latent space $Z_0$. Since we use adversarial training, we define two Adam optimisers [4] with a learning rate of $0.0001$ and a batch size of 60 that optimise our objective on an alternating basis. For better visualisation, we discretise $X$ and $Y$ into 10 bins to colour-code to show the invariant parts of the data. We set $\beta$ to 1 such that we have approximately the same weight on the loss terms. To obtain the optimal number of latent dimension, we used the same procedure as in Additional Experiment 2.

**STIB without adversary** For the STIB without adversary setup, we use exactly the same configuration as in the STIB setup. The only difference is that we remove the adversarial mutual information regulariser. Thus, we define only one Adam optimiser with a learning rate of $0.0001$ and a batch size of 60 which optimises our loss function.

**VAE** This VAE setup uses also the same configuration as STIB. Similar to STIB without adversary, we skip the mutual information regulariser. In addition, we only define a shared latent space $Z$ to reconstruct $X$ and $Y$ in contrast to STIB. As an optimiser, we employ Adam with a learning rate of $0.0001$ and a batch size of 60.

**CVAE** For CVAE, we use the same setup for encoder and decoder as in STIB. In contrast, we model the latent space $Z$ with two dimensions by estimating the parameters $\mu$ and $\sigma$. We employ a two-dimensional latent space as we concatenate the one-dimensional part in traditional CVAE fashion. In order to train our model, we use Adam with a learning rate of $0.0001$ and a batch size of 60.

**CVIB** For the last model, we use the equivalent model as in CVAE. In addition, we employ the mutual information regulariser which is described in [9].

### 2.1.2 Additional Experiment 1:

In the additional experiment, we qualitatively inspect the ability of the latent space to approximately reconstruct $X$ and $Y$ from our latent representation. The reconstruction of $X$ is depicted in Figure 1 and the reconstruction of $Y$ in Figure 2. The color coded lines in the Figures 1 and 2 indicate the invariant parts in the dataset. Note in our artificial examples the invariances are continuous. However, we discretised the invariances into 10 colour-coded bins for visualisation proposes. In the first part, we compare our input $X$ in Fig. 1a with the reconstruction ability of the models, namely STIB (Fig. 1b), VAE (Fig. 1c), STIB without adversary (Fig. 1d), CVAE (Fig. 1e) and CVIB (Fig. 1f). First, we inspect the ability to reconstruct the input $X$. From a visual perspective STIB, VAE, STIB without adversary, CVAE are approximately able to reconstruct the input $X$. Except CVIB can only partially reconstruct $X$. These qualitative support the quantitative findings which we have obtained in Experiment 2 in the main paper. In this experiment, we investigated the reconstruction of $X$ by quantitatively evaluating the MAE.

In the second part, we examine the reconstruction capability of $Y$ (Fig. 2). Here, we do not compare to CVAE and CVIB because in these particular models $Y$ is used as an input and not reconstructed by the model. In comparison to the ground truth (Fig. 2a), STIB in Figure 2b is able to reconstruct the backbone of the spiral. We cannot reconstruct the $Y$ in detail, because we draw noisy data points of $Y$ (for more details see Dataset). In contrast, VAE (see Fig 2c) uses three latent dimensions $Z$ instead of one. Therefore, VAE not only tries to reconstruct the spiral but also tries to learn the noise of the data. This is clearly indicated by the noisy reconstruction of the $Y$. Last, we compare to STIB without adversary in Figure 2d. Similarly to STIB, we are able to reconstruct $Y$ which also confirms the quantitative findings from Experiment 2 in the main paper.

Figure 1: The first image denotes the input $X$ (Fig. 1a) whereas the second image (Fig. 1b) illustrates the reconstruction of STIB. Images three (Fig 1c) and four (Fig. 1d) denotes the reconstruction results of VAE and STIB without adversary, respectively. In Figure 1e, we show the reconstruction of CVAE and in Figure 1f the results of CVIB. For better visualisation, we discretise $X$ into 10 bins to colour-code to show which part of the data is invariant.

Figure 2: The first image illustrates the output $Y$ (Fig. 2a). The second image (Fig. 2b), illustrates the reconstruction results of STIB whereas the third column shows the VAE reconstruction of $Y$ (Fig. 2c). The last column (Fig. 2d) shows the results for STIB without mutual information regulariser. We have not included results for CVAE and CVIB because $Y$ is not reconstructed but used as an additional input. To better showcase which parts of the data is invariant, we discretise $Y$ into 10 colour-coded bins.

## 2.2 Real Experiment: QM9

### 2.2.1 Experimental Setup:

**STIB** As input $X$ we use the SMILES representation of a molecule, which encodes molecular connectivity in a string based graph and chemical properties as target $Y$. In doing so, we are converting the SMILES in a one-hot grammar representation based on the Grammar VAE introduced by Kusner *et al*. [7]. As proposed in Kusner *et al*. [7], our encoder network consists of three 1D convolutional layers with 12 convolutional filters and a filter length of 3, followed by a 256 dimensional fully connected layer. In addition, we have two decoders that try to reconstruct both the chemical properties and the SMILES. The SMILES decoder consists of a 36 dimensional fully connected layer followed by three 256 dimensional Gated Recurrent Unit (GRU) layers. The properties decoder has four fully connected layers with 56, 256, 128, 2 nodes, respectively which is similar to [3]. Last, we define the adversarial network to minimize the mutual information between $Z_0$ and $Y$ with the same configuration as the property decoder. Our model is trained using two Adam optimisers [4] with an initial learning rate of $0.01$ and a batch size of 36. Subsequently, we set our latent dimension $Z$ to 16 because the reconstruction accuracy saturates (see Fig. 5 main paper). We split $Z$ to $Z_0$ with 14 and $Z_1$ with 2 dimensions because we predict two target properties. To speedup the training procedure, we pretrained the encoder and decoder on approximately 100k molecules from the QM9 dataset and subsequently fine-tuned the latent representation on the fixed stoichiometry ($C_7O_2H_{10}$). We set $\beta$ to 1 such that we have approximately the same weight on the loss terms.

**STIB without adversary** Here, we use the same configuration as in the STIB setup. However, we omit the adversary with the mutual information regulariser. As we do not have an adversary, our model uses only one Adam optimiser with an initial learning rate of $0.01$ and a batch size of 36.

**VAE** In the VAE setup, we employ the equivalent architecture as in STIB without adversary. However, we do not partition $Z$ into two separate latent spaces. Instead, we reconstruct $X$ and $Y$ from the $Z$ directly. In this case, we devote Adam with an initial learning rate of $0.01$ and a batch size of 36.

**CVAE** For the CVAE setup, we take the both encoder and decoder architecture of VAE. However, we do no reconstruct $Y$ but concatenate $Y$ with the latent representation $Z$ in order to reconstruct $X$.

Thus, our latent space has only a latent dimensionality of 14 instead of 16 in the VAE architecture. Similar to the models described before, we train CVAE using the Adam optimiser with an initial learning rate of $0.01$ and a batch size of 36.

**CVIB** For CVIB, we use exactly the same setup as for CVAE. The only difference is that we add the mutual information regulariser, developed in [9] to the CVAE loss function.

**Additional Experiment 2.** We inspect the molecule reconstruction ability of the input $X$ given a varying number of latent dimensions (Fig. 3). To do so, we train our model on $95\%$ of our dataset and subsequently evaluate on the remaining $5\%$. The model selection is hence performed by inspecting the reconstruction accuracy to select the optimal number of latent dimensions. In our case, the optimal model converges at 16 latent dimensions. Reconstructing molecules from lower dimensions is in general more challenging because there is a large number of molecules with similar bandgap energies and polarisability. This results in collisions which makes it difficult to resolve the many-to-one mapping in the latent space. In addition, we calculated the mutual information between $Z_0$ and $Y$ using the Kraskov estimator. It is important to note that our model does not come with a trade-off between the reconstruction accuracy and being invariant against $Y$ in $Z_0$. This property is clearly indicated in Figure 3 (blue line). Here, it can be observed that the mutual information constantly stays between 0.03 and 0.1 for all numbers of latent dimensions considered.

Figure 3: Illustration of the model selection process of STIB on the testset defined in Experiment 4. Therefore, the SMILES reconstruction accuracy (green dot) is considered. The x-axis denotes the number of latent dimensions whereas the left y-axis depicts the reconstruction accuracy of the molecules. The plot indicates that our reconstruction rate saturates at a level of 99% even when varying the number of latent dimensions. In addition, we plotted the mutual information (blue cross) between $Z_0$ and $Y$ for all models which is depicted by the right y-axis.

## 2.3 Real Experiment: Zinc Dataset

**Dataset.** In the third experiment, we use the 250K drug-like molecules from the ZINC database [3]. In contrast to QM9, this dataset consists of up to 23 heavy-atoms (C, O, N and F), not including hydrogens and offers a larger variety of molecule structures. The dataset is a randomly picked subset of the larger ZINC database [14] which contains over 17 million molecules. Here, every molecule has calculated drug-specific properties such as synthetic accessibility score(SAS) or the Qualitative Estimate of Drug-likeness (QED).

### 2.3.1 Experimental Setup:

**STIB** As input $X$ we use the SMILES representation of a molecule as in the QM9 experiment. In contrast to the previous experiment, we are converting the SMILES in a one-hot representation based on DeepSMILES [10] instead of the grammar representation by Kusner et al. [7]. DeepSMILES preprocesses a given SMILES string to a simpler representation which can be easier learned by recurrent neural networks. For our encoder network, we use one GRU layer with hidden 12 dimensions followed by two fully connected layers with 1356 and 128 dimensions, respectively. Subsequently, we set our latent dimension $Z$ to 20 and split it to $Z_0$ with 19 and $Z_1$ with 1 dimension because we predict only the drug-likeliness of the molecules. Last, we define the decoder networks. Therefore, we have a SMILES decoder which consists of a 36 dimensional fully connected layer followed by three 501 dimensional Gated Recurrent Unit (GRU) layers. In addition, we define a property

decoder which consists of four fully connected layers with 36, 36 and 1 nodes, respectively. The adversary decoder, which is responsible to minimise the mutual information between $Z_0$ and $Y$ has got three layers with 36, 36 and 1 nodes, respectively. At the end, the model is trained using the Adam optimizer with an initial learning rate of $0.001$ and a batch size of 500. We set $\beta$ to 1 such that we have approximately the same weight on the loss terms.

**STIB without adversary** As stated in the two experiments before, we use the same architecture as in STIB. However, we leave out its adversarial part. Thus, we employ only one Adam optimiser with an initial learning rate of $0.001$ and a batch size of 500.

**VAE** Here, we make use of the same encoder/decoder architecture as in STIB without adversary. However, instead of splitting $Z$ into two separate latent spaces $Z_0$ and $Z_1$, we reconstruct $X$ and $Y$ from $Z$.

**CVAE** We employ the the same setup as for the VAE. However, we do predict $Y$ from $Z$ but concatenate $Y$ with the latent representation $Z$. For this reason, we set the dimensionality of $Z$ 14 dimensions in contrast to 16 dimensions in the VAE architecture. Again, we use Adam to train our model with an initial learning rate of $0.01$ and a batch size of 36.

**CVIB** Here, we use the identical setup as for CVAE. The merely distinction is the mutual information regulariser introduced in [9] which is added to the CVAE loss function.

**Additional Experiment 3.** In this experiment, we perform an additional quantitative study on Zinc dataset (Table 1). Here, we also obain competitive reconstruction results in terms of SMILES accuracy and druglikeliness. For all models, we received a SMILES reconstruction accuracy of 98% for VAE, 98%, for STIB without adversarial training scheme, 98% for CVAE, 94% for CVIB and 98% for STIB. Furthermore, we investigated the druglikeliness MAE where all models received 0.05. This shows that our approach receives competitive results in both reconstruction tasks in comparison the baseline. Last, we estimated the mutual information with a Kraskov estimator between the target-invariant space $Z_0$ and the target $Y$. The VAE baseline contains 0.80 bits mutual information whereas STIB without adversary contains 0.24 bits. Moreover, we received a mutual information on 0.28 bits and 0.29 bits for CVAE and CVIB, respectively. That implies that all considered models contain mutual information in $Z_0$ about $Y$ whereas if we employ STIB the mutual information is approximately eliminated (0.07 bits). These results confirm the findings of Experiment 2 and 5 that only STIB is able learn symmetry transformations from data while archiving competitive reconstruction results.

**Additional Experiment 4.** Lastly, we investigate the generative nature and investigate the property consistency of our model. To do so, we fix three different points in property-latent space $Z_1$. The points in property-latent spaces represent a druglikeliness of 0.5, 0.7, 0.9, for rows one to three in Figure 4a, respectively. After, we randomly sample points in the invariant latent space $Z_0$ which are subsequently generated to SMILES strings.

(a)                    (b)

Figure 4: Illustration of the generative process of our model. Figure 4a shows samples drawn by our model. The labels represent the predicted druglikeliness properties which were estimated by out model. Each row in Figure 4a denotes molecules generated with a predefined druglikeliness. We further estimate the properties of the generated molecules and show the result in Figure 4b. The blue shaded background is the error confidence interval of our model and the x-axis denotes the MAE of all samples in the boxplot.

Having generated novel SMILES with potentially identical druglikeliness, we now perform a self-consistency check. That is, we feed the generated SMILES into our model and predict the properties.

If our model has learned an invariant representation the predicted druglikeliness should be identical to the fixed druglikeliness within the model error. We summarised the results of the model-consitency check in Figure 4b. Here, we plot the predicted druglikeliness averaged over all generated molecules from the three reference points using a boxplot. Every boxplot contains between 108 and 193 sampled molecules. The x-axis denotes the druglikeliness MAE whereas the blue box denotes the model error. The predicted properties averaged over all generated molecules from the three reference points posses a MAE between 0.04 and 0.05 which lies within calculated the model error range in Table 1. This observation is additionally supported by investigating the boxplots. Here, the predominant proportion of molecules lie within the model error range (blue box). Hence, this experiment demonstrates the generative capabilities of STIB by generating chemically correct novel molecules within the model's error range.

Table 1: Summary of quantitative results for Zinc experiment. Here, we consider VAE, STIB without regularization, CVAE, CVIB and STIB. The accuracy for SMILES and MAE reconstruction error for druglikeliness (probability) are computed, as well as the mutual information (bits) between the invariant space $Z_0$ and $Y$ based on a Kraskov estimator ($\text{MI}_\text{K}(Z_0,Y)$). Higher SMILES accuracy and lower MAE and MI are better. STIB outperforms the other baselines.

| MODEL | ZINC | | |
|---|---|---|---|
| | SMILES | DRUGLIKELINESS | $\text{MI}_\text{K}(Z_0,Y)$ |
| VAE | **0.98** | **0.05** | 0.80 |
| STIB W/O ADV. | **0.98** | **0.05** | 0.24 |
| CONDVAE | **0.98** | - | 0.28 |
| CVIB | 0.94 | - | 0.29 |
| STIB | **0.98** | **0.05** | **0.07** |