[Reviews · NeurIPS 2020]

Review 1

Summary and Contributions: The authors propose a model based on the variational information bottleneck, with the addition of splitting the latent variable into two components that capture distinct information about two components. They offer a new adversarial objective to enforce this property, and run several experiments on synthetic and molecule datasets.

Strengths: The motivation is clear, and the suggested objective seems reasonable. In particular the solution of parameterizing an observation as Gaussian but calculating the mutual information through a bijective transformation seems novel and sound.

Weaknesses: The results on the QM9 dataset seem weak, in that the proposed model clearly minimizes mutual information but performs comparably to a regular VAE on the more applicable metrics. I understand that minimizing the mutual information is the intended aim of the model, but I'm not sure I see the motivation for optimizing that measure in isolation. In particular, the end of section 2 mentions this paper is more interested in a generative model rather than learning a latent space to be used for downstream tasks. In that case, the paper might be strengthened by using that generative model sampling and studying the outputs.

Correctness: The claims and methodology appear correct.

Clarity: The paper is generally well-written. Notationally, I found the mutual information terms somewhat subtle, as the subscript \phi and \theta obscure which conditional distributions are being learned in each term. Also, on page 5, what is meant by the covariance of Z_0Y?

Relation to Prior Work: The comparison to other models in this vein seems thorough.

Reproducibility: Yes

Additional Feedback: Post-Rebuttal: I've read the author responses, I believe my score is still appropriate.


Review 2

Summary and Contributions: The paper proposes a method to learn a latent space representation of quantities/observables that are of interest, are nonlinear functions of a complex input such as a molecular graph, and we would like to have them in an ordered/organized way. The authors look this learning problem through the lens of symmetry groups.

Strengths: The paper proposes an interesting and novel idea to approach symmetries via learning an encoding space wrt some target quantity which is kept as an invariant. The authors illustrate the concept on a simple toy example and on a molecular dataset, where learning the space which "orders" the data wrt a complex nonlinear quantity (such as an energy or band gap across chemical space) is very challenging. In principle such a method could be very useful in molecular design, and probably also in AI-driven design tasks in other areas. The idea is to organize the training data such that they are order wrt quantities of interest in the latent space, and then search new molecules (or other objects to be optimized) by making modifications on them, projecting them into this ordered latent space and then pushing the optimization along the quantity to be optimized, e.g. within a RL framework.

Weaknesses: The paper has some serious drawbacks that are unlikely to be solved by a modified paper and rather require more significant changes. - The discussion of the difficulty of estimating MI in the experiments is not given. As this goes to the core of evaluating the method in contrast to competitors this is critical. - It it is absolutely unclear why h, h^{-1] should form a bijection and its inverse given this training objective. It is furthermore unclear why this should lead to Gaussianization of \tilde Y. It this remains unclear what (8) and (9) are actually measuring and thus what the adversarial training approach is optimizing. - The point of doubt of the last bullet has to be transferred to the experiments. If Z0 should be the space of non-trivial changes of structures with same target measure Y, it would be critically important to see what these learned changes consist of or what Z0 actually represents. Right now Z0 could just be irrelevant information w.r.t. target and e.g. figure 2 in the appendix indicates that reconstructed samples are only supported in 1D and thus could be completely explained by Z1 alone. It is still unclear what "symmetry" means here and whether it can be learned from data alone without other principled prior information e.g. as derived from domain knowledge. Right now points are mapped to a target-invariant code space. Symmetry transformations could be defined by mapping a molecule into this code space, modifying the code, and then moving back into conformation space. However, this paper does not show that this leads to meaningful transformations that are chemically sensible. Overall, my feeling is that looking at this problem through the lens of symmetry is not helpful at all. When considering symmetries one usually has certain geometric operations in mind (such as the rotation cited in Fig. 1a), which invoke a symmetry group with certain properties, and then one can consider to develop NN architectures that preserve these symmetries or are equivariant etc. But none of the properties arising from such a treatment are exploited here. This paper is about learning structured latent spaces, and it would probably serve the authors better to discuss it this way.

Correctness: For some claims it is unclear if the numerical evidence supports them. Experiment 2: How do you show that Z0 is even necessary for reconstruction now? The data set appears to be 1D. In the appendix you can also see that reconstructions are structured on a 1D curve embedded in the 2D space. There is barely any variance in the normal space of the curve. Couldn't this just indicate that Z0 is completely ignored for reconstruction (then having MI of 0 isn't so impressive) while all reconstructive power is left in Z1 (including the entanglement with the target)? Then using a highly complex decoder like an NN could embed the variation of Z1 as an embedded 1D curve explaining the reconstruction results. Note that the VAE results, while more blurry regarding the true curve, indeed shows variance in off the curve. One should prove that this is not the case using some ablative study - e.g. seeing the influene of manipulating Z0 on the resulting samples. Especially if Z0 should serve as a representation space for interesting and non-trivial symmetry transformations, you would need to show that this does not represent a trivial set of transformations e.g. the identity + minimal noise. L 274-276: In order to make such claim you need to show an ablative study on what influence Z0 actually has on generated structures. Does it really produce new structures? Which role does it play in reconstruction? How do you guarantee that the model isn't just pushing all information into Z1 so Z0 does not really contribute to structural changes? As long as those questions remain unanswered in this main text the whole point of the approach is not really proven. MI estimation: The Kraskov estimator can perform inferior when MI among two variables is high. Also estimating MI gets increasingly harder for more dimensions. It also servers as a LOWER bound. This means all given quantities could in reality be higher than shown in the tables.

Clarity: I think the symmetry view on the problem is really confusing and misleading, and this makes the paper hard to read IMO. Also the description of the results and what they really demonstrate could be improved (see above).

Relation to Prior Work: Yes. Here are some citations relevant for the MI estimation: [1] Gao, S. et. al, Efficient estimation of mutual information for strongly dependent variables. AISTATS 2015 [2] McAllester et. al, Formal Limitations on the Measurement of Mutual Information, AISTATS 2020

Reproducibility: Yes

Additional Feedback: Post-review: in the light of the authors response I have raised my score from 3 to 4. However, my more general reservations remain and I think it's a weak concept and weak paper, although I do not dispute that symmetries in ML are important in general. Thereby I maintain the opinion it should be rejected from NeurIPS.


Review 3

Summary and Contributions: This paper proposes STIB to learn symmetry transformation using adversarial training and a partitioned latent space. They also introduce a continuous mutual information regulation approach to handle one-to-one transformations in the continuous domain. Experiments show that this method achieves state-of-the-arts.

Strengths: 1) The problem focused on this paper is very interesting. 2) The method is well modeled in a conceptually intuitive way 3) Extensive experiments demonstrate the effectiveness of the proposed method

Weaknesses: 1) Why symmetry is very important? 2) In deep learning, especially in practical tasks, e.g., classification etc., does symmetry affect the performance of neural networks? 3) The training mechanism is quite common. In my opinion, this work is a slightly-modified version of GAN+VAE framework. Please illustrate more insightful contents or the major differences

Correctness: Yes

Clarity: Yes

Relation to Prior Work: Moderately clear

Reproducibility: Yes

Additional Feedback: UPDATE: After reading the rebuttal, only part of concerns have been resolved. So, I still keep my point.


Review 4

Summary and Contributions: The paper proposes to use deep latent-variable models to directly learn from observations the symmetric transformation, which are frequently observed in the chemical and physics spaces. Technically, the paper mainly studies the problem of disentangled feature learning. Similar to the previous works on this topic, the paper also formulates the problem under the framework of deep information bottleneck theory, and then apply the adversarial training technique to train the model, forcing the mutual information between latent representations and the original data X and target Y moving towards the desired direction. From a technical perspective, the main contribution of this paper is that they proposed a method to estimate the mutual information when the target Y is continuous, by assuming the Gaussian distribution for Y. The Gaussian assumption can be relaxed by using a learned bijective mapping. An artificial and two real-world datasets are used in the experiments to verify the performance of the proposed method.

Strengths: 1) Although it's hard for a non-biologist to fully understand the significance of learning a symmetric transformation in chemical from data automatically, it has convinced me it is an important problem, and is also very interesting. 2) The paper considers the problem of disentangled representation learning when the target variable is continuous, in contrast to the most previous works in which Y is assumed discrete.

Weaknesses: My main concern on this work is its technical depth. As I said in the summary, the problem raised in this paper is very interesting, and potentially important. But from a technical perspective, what the paper has really done is the disentangled representation learning. This problem is tackled with a very similar approach as previous works, i.e., formulating the problem under deep information bottleneck, and then training it with adversary techniques. The main contribution is the newly proposed method to estimate the mutual information between the target Y and latent representation under the scenario where Y is continuous. The estimation method is somewhat standard, and is not interesting enough. From the perspective of technical depth, I don’t think the paper deserves to appear in NeurIPS. Secondly, although the paper did a lot of experiments, one on artificial dataset and two on real-world datasets, it does not directly present to us the benefits brought by the methods in a way that can be understood by ordinal people. Thirdly, there should be more explanation about the technique of relaxing Gaussian assumption with bijective mapping. In the submitted codes, you calculate the bijective loss with || h^{-1}(h(Y)) - h(Y) ||, which is different from what you defined in the paper || h^{-1}(h(Y)) - Y || Lastly, the paper aims to study the problem of symmetric transformation learning, but it mostly talks about disentangled representation learning, except the motivation in the introduction and the datasets in the experiments. Essentially, the paper didn’t talk too much about the symmetric transformation learning. I guess the authors may consider restructuring the paper, e.g., writing it from the perspective of disentangled representation learning, and using the symmetric transformation as an application.

Correctness: mostly correct

Clarity: yes

Relation to Prior Work: nil

Reproducibility: Yes

Additional Feedback:


Review 5

Summary and Contributions: A variational auto encoder is trained with an adversarial network to disentangle the latent subspace representing an attribute from the latent subspace representing an instance by minimizing the mutual information between the attribute and the instance latent subspace. A second contribution is in the computation of the mutual information as a Gaussian distribution transformed by a bijection, which allows the use of correlation matrices. Experiments are shown on molecular datasets and toy problems.

Strengths: I like this paper. The application of disentanglement to discover symmetries with respect to molecular bandgap energies is eye opening and the method appears sound. The experimental results are superior to prior works.

Weaknesses: The paper may benefit from relating the language a little more to the parlance of the literature. For example, symmetry transformations arise from disentangled latent representations, but the term "disentangle" appears only in some titles in the References section. The authors reiterate their contribution of handling continuous variables, but appear to omit prior works that compute disentangled continuous latent spaces, even some using adversarial networks and mutual information, though it appears using a different formulation (see section on prior work below). A section that ties all of these ideas together would strengthen the paper, especially considering that this is named as the primary contribution of the paper. It's not clear what advantage the bijection and correlation matrix idea has over established methods using normalizing flows to implement more expressive continuous posteriors (also noted in the section on prior work below). Since this is listed as the second contribution of the paper, it would be appropriate to make such comparisons.

Correctness: The paper appears correct as far as I could tell, and the experiments seem sound.

Clarity: The paper language is very clear, other than my desire to see the authors relate their language to the disentanglement literature which may allow readers familiar with this literature to grasp the paper more quickly. There are a few places where the main points of the paper are repeated, more than twice, across various sections. This could be tidied up but it's not bad. The experiments are plotted very clearly and we can observe the invariances directly. That's nice.

Relation to Prior Work: There are several prior works that disentangle continuous latent subspaces using adversarial networks and mutual information objectives. It may benefit the community to comment on the similarities and differences with these prior works. For example, comparing Eq. 4 in the author's paper vs Eqs. 3 and 6 in Chen et al. "InfoGAN: Interpretable Representation Learning by Information Maximizing Generative Adversarial Nets", 2016. It would also be interesting to consider prior works that employ subspace swapping, such as Matheiu et al, "Disentangling factors of variation in deep representations using adversarial training", 2016. The bijection employed in Fig. 3 is essentially an invertible network, and it seems obvious to apply the vast literature on normalizing flows to this task. Invertible layers may be used to avoid learning both directions of the mapping, and the normalizing flow may be employed as a more expressive posterior, much like what the authors are attempting (as in Rezende and Mohamed, "Variational Inference with Normalizing Flows", 2015 or Kingma et al, "Improved Variational Inference with Inverse Autoregressive Flow", 2016). There is also an entire conference on this topic (INNF - ICML Workshop on Invertible Neural Networks, Normalizing Flows, and Explicit Likelihood Models).

Reproducibility: Yes

Additional Feedback: The authors may be interested in an analogous problem in computer graphics and computer vision: encoding faces with disentangled latent subspaces representing identity and expression. In this case, Z_0 is analogous to identity, and Z_1 is analogous to expression. There is vast literature on training strategies for this disentanglement, and I encourage authors to cross over to other application disciplines when researching literature or when looking for benchmarks to run their algorithms on. After the author feedback cycle and reading the other reviews, I get the impression that my score calibration may be off for papers in this area. However I'll retain my overall score for fairness, but lower my confidence score to reflect this impression.


Review 6

Summary and Contributions: This paper proposed to learn the symmetry transformation with a model consisting of two latent subspaces Z0 and Z1, where the first subspace Z1 contains information about input and target, while the second subspace Z0 is invariant to the target. The central element is the proposed Symmetry-Transformation Information Bottleneck (STIB) model that learns a continuous low-dimensional representation of the input data and the corresponding symmetry transformation in an inverse fashion. The authors focus on minimizing mutual information in continuous domain, so the calculation is based on mutual information on the correlation matrices in combination with a bijective variable transformation. The model effectiveness is demonstrated by state of the art performance on both artificial and real-world molecular datasets.

Strengths: The problem is well motivated, backed by detailed analysis in the preliminary sections. The main technical contribution is well derived in Section 4, together with relaxations of the Gaussian assumption of the method, and the effectiveness is verified by artificial and real-world experiments in Section 5. As an algorithm paper, I think the experiments are sufficient to prove the claim. Based on the evaluation, the proposed Symmetry-Invariant Information Bottleneck can benefit the NeurIPS community.

Weaknesses: The paper is good overall. There is no obvious weakness.

Correctness: The claims and method is correct. I do not find obvious inconsistency upon review.

Clarity: Yes, the paper is well written and easy to follow.

Relation to Prior Work: The proposed method is closely related to Information Bottleneck (IB) method and fairness topics. The authors have discussed the relations and differences in Section 2.

Reproducibility: Yes

Additional Feedback: Post rebuttal: the authors have answered most of the concerns raised by the reviewers in my opinion. I think this is still a good paper to be accepted to NeurIPS. I will maintain my original rating.

[Author Response · NeurIPS 2020]

We are truly appreciative to all reviewers for their insightful and very helpful comments. Overall, we propose a novel mutual information (MI) regularisation method to remove continuous target-information from latent representations. We believe our work should be shared with the community as it demonstrates the effectiveness of our method and has numerous socially-relevant applications such as drug discovery and solar cell design. We emphasise that moving from discrete to continuous targets is not straightforward as minimising MI in such settings presents several difficulties. Thus, we view our method as a novel solution to a challenging problem. Moreover, the reviewers acknowledged that the paper is "well written and easy to follow" (R1, R8), "very useful in molecular design" (R2), "novel and interesting" (R1, R2, R4, R3), "well motivated" (R4,R8) and the "method appears sound" (R1,R5). We address the reviewer comments below:

(R1, R2) **Claim ll. 274–276, what influence $Z_0$ actually has on generated structures?, Paper strengthened by sampling from generative model.** We agree that this is a highly important and relevant question. Firstly, it is not possible to capture all information in the 2-dim. $Z_1$ since we require at least 16 dim. for a good reconstruction (see Appendix, Exp. 2 + Fig. 3) Secondly, we performed generative experiments (see Add. Exp. 4) on the highly complex Zinc dataset (>250k molecules) to demonstrate the effectiveness of our approach. We will add the same experiments for QM9 to the appendix in the final version.

(R2) **The Kraskov estimator can perform inferior when MI is high, Estimate only lower bound.** This remark is true, however, in the regime we consider (minimising the MI), there is almost no dependence between $Z_0$ and $Y$. In addition, we demonstrated that the MI is small by looking at the qualitative results of the latent space (e.g. Fig. 4) and at the generative nature of our model (Appendix, Add. Exp. 4).

(R2) **is $Z_0$ even necessary for reconstruction now?, Reconstructed samples are only supported in 1D and thus could be completely explained by $Z_1$ alone?** This is a misunderstanding. The dataset is constructed such that every point on the diagonal ($X$) maps to the same point in $Y$. Therefore, we need additional dimensions to reconstruct the position on the diagonal. We performed an additional experiment where $Z$ is only 1D. This leads to a MAE(X)=1.97 and a MAE(Y)=0.67 which indicates that a 1D space is not sufficient to reconstruct $X$.

(R2, R4, R5) **paper aims to study symmetric transformation learning, but it mostly talks about disentangled representation learning, unclear what "symmetry" means here, When considering symmetries one usually has certain geometric operations in mind (such as the rotation cited in Fig. 1a).** As we state in the introduction (line 33), the goal of the paper is to learn a symmetry property $f$ of the system that leads to a predefined invariance ($Y$). The purpose of our model, however, is to go beyond simple geometric operations and to allow for learning arbitrary continuous transformations that result in the invariance. In general, by considering arbitrary continuous transformations $g$ (Fig. 1), we model the group action of a Lie group (the set of $g$) on the space $X$ that preserves the symmetry $f$. We will also extend the related work section with a survey of related disentanglement approaches.

(R2, R4, R5) **It it is absolutely unclear why $h$, $h^{-1}$ should form a bijection, there should be more explanation about the technique of relaxing Gaussian assumption with bijective mapping, The bijection employed in Fig. 3 is essentially an invertible network.** Since both $h$ and $h^{-1}$ are functions between continuous sets, the loss given in line 181 can only be 0 if both functions form a one-to-one mapping. Thus, Eqs. (8,9) do measure the actual mutual information. This is indeed also a feature of an invertible network and using one is a valid alternative the relaxation technique we employed.

(R4) **In the submitted codes, you calculate the bijective loss with $||h^{-1}(h(Y)) - h(Y)||$, which is different from what you defined in the paper $||h^{-1}(h(Y)) - Y||$.** The correct equation is $||h^{-1}(h(Y)) - Y||$. We uploaded the wrong code, the corrected results with the loss in the paper are: MAE(X)=0.05, MAE(Y)=0.44, MI(Z0,Y)=0.19. The results in the real experiments are not affected as the property data is approx. Gaussian which is why we have not used the bijection extension.

(R3, R4) **Why symmetry is very important? What is the benefit of the method for chemistry?** In material science, e.g. solar cell design, we want to find all variations of molecules that posses the same bandgap energy of 1.2 eV to adequately generate electricity. Therefore, we need to find a transformation that alters a molecule and leaves the property unchanged (see ll. 22–32).

(R3, R4) **this work is a slightly-modified version of GAN+VAE framework. Please illustrate more insightful contents or the major differences.** Moving from discrete to continuous targets is not straightforward as minimising MI in such settings gives rise to several difficulties. To the best of our knowledge, cognate models have solely focused on discrete $Y$. This is because naively using the negative log-likelihood (NLL) as done when maximising mutual information in other deep information bottleneck models leads to critical problems in continuous domains. This stems from the fact that fundamental properties of mutual information, such as invariance to one-to-one transformations, are not captured by this mutual information estimator. Moreover, we want to consider multiple properties at once, where every one requires high resolution. Simultaneous high-resolutional discretisation of multiple targets would result in an intractable classification problem.

(R2) **discussion of the difficulty of estimating MI in the experiments is not given.** Throughout our model, we use the analytic formula for Gaussian MI (Eqs. (8,9)) which we extend with the Gaussian relaxation. We subsequently use the Kraskov estimator as a benchmark. A comparison to different approaches of MI estimation such as MINE is not a focal point of the paper, but we will add a short discussion of suggested related methods in case of acceptance.

[Meta-Review · NeurIPS 2020]

There is quite some disagreement between the reviewers, but at the same time there is agreement that the paper introduces an interesting method which does symmetry transformation by splitting the latent space. The rebuttal removes many doubts of the reviewers, and even if not all issues are completely cleared I venture an accept.